# Human Defensins: A Novel Approach in the Fight against Skin Colonizing *Staphylococcus aureus*

**DOI:** 10.3390/antibiotics9040198

**Published:** 2020-04-21

**Authors:** Olga Scudiero, Mariarita Brancaccio, Cristina Mennitti, Sonia Laneri, Barbara Lombardo, Margherita G. De Biasi, Eliana De Gregorio, Chiara Pagliuca, Roberta Colicchio, Paola Salvatore, Raffaela Pero

**Affiliations:** 1Department of Molecular Medicine and Medical Biotechnology, University of Naples Federico II, Via S. Pansini 5, 80131 Naples, Italy; cristina.mennitti@studenti.unina.it (C.M.); barbara.lombardo@unina.it (B.L.); edegrego@unina.it (E.D.G.); chiara.pagliuca@unina.it (C.P.); roberta.colicchio@unina.it (R.C.); paola.salvatore@unina.it (P.S.); 2Ceinge Biotecnologie Avanzate S. C. a R. L., 80131 Naples, Italy; 3Task Force on Microbiome Studies, University of Naples Federico II, 80100 Naples, Italy; 4Department of Biology and Evolution of Marine Organisms, Stazione Zoologica Anton Dohrn, Villa Comunale, 80121 Naples, Italy; mariarita.brancaccio@szn.it; 5Department of Pharmacy, University of Naples Federico II, 80131 Naples, Italy; slaneri@unina.it (S.L.); margherita.debiasi@unina.it (M.G.D.B.)

**Keywords:** skin infection, *S. aureus*, human defensins

## Abstract

*Staphylococcus aureus* is a microorganism capable of causing numerous diseases of the human skin. The incidence of *S. aureus* skin infections reflects the conflict between the host skin′s immune defenses and the *S. aureus’* virulence elements. Antimicrobial peptides (AMPs) are small protein molecules involved in numerous biological activities, playing a very important role in the innate immunity. They constitute the defense of the host′s skin, which prevents harmful microorganisms from entering the epithelial barrier, including *S. aureus.* However, *S. aureus* uses ambiguous mechanisms against host defenses by promoting colonization and skin infections. Our review aims to provide a reference collection on host-pathogen interactions in skin disorders, including *S. aureus* infections and its resistance to methicillin (MRSA). In addition to these, we discuss the involvement of defensins and other innate immunity mediators (i.e., toll receptors, interleukin-1, and interleukin-17), involved in the defense of the host against the skin disorders caused by *S. aureus*, and then focus on the evasion mechanisms developed by the pathogenic microorganism under analysis. This review provides the “state of the art” on molecular mechanisms underlying *S. aureus* skin infection and the pharmacological potential of AMPs as a new therapeutic strategy, in order to define alternative directions in the fight against cutaneous disease.

## 1. Introduction

*Staphylococcus aureus* is a Gram-positive bacterium. It shows opportunistic pathogen activity in both humans and animals. In fact, it can cause multiple disorders such as infections or abscesses of the skin, endocarditis, sepsis, urinary tract infections (UTI), mastitis, meningitis, osteomyelitis, food poisoning, and infections associated with biofilm and septicemia [1,2,3,4] (Figure 1). *S. aureus* is a common component of human skin microbiota and it colonizes the nasal mucosa of the world’s population. Furthermore, *S. aureus* releases staphylococcal enterotoxin A (SEA) into food, and this enterotoxin causes food-borne poisoning. Moreover, on the one hand, SEA is involved in nosocomial infections, which occur in about 30% of the cases of infective endocarditis [5,6]; on the other hand, it represents a major cause of hospital-acquired pneumonia [7,8]. Furthermore, *S. aureus* colonization has a negative impact on some chronic inflammatory dermatoses—in particular, atopic dermatitis (AD), a multifactorial complex disease that causes skin barrier dysfunction [9,10].

The pathogenicity of *S. aureus* is due to a series of virulence factors, such as enterotoxins, exfoliative toxins, and Panton-Valentine leukocidin (PVL) [11,12]; consequently, it can cause diseases in healthy individuals due to the expression of these virulence factors [1,12]. Moreover, it can lead to the formation of biofilms on tissues and devices implanted at a medical level [11]; these characteristics allow *S. aureus* to invade the tissues and to spread, leading to systemic diseases. *S. aureus* has developed different strains that show methicillin resistance (MRSA) or multi-drug resistance (MDR) [13]. Generally, mecA and mecC, located on the staphylococcal chromosome cassette mec (SCCmec), are involved in methicillin resistance, causing broad-spectrum resistance against all β-lactam antibiotics [13,14,15]. In contrast, MDR is described as an acquired resistance to three or more classes of antibiotics; however, in some cases, a number of *S. aureus* strains exhibit resistance against all commonly used antibiotics [16]. Therefore, finding alternative treatments for MRSA or MDR infections is a worldwide public health priority [17]. At the same time, it is necessary to improve biosecurity on farms with a concomitant development of vaccines to optimize the control of *S. aureus* infections in animals [18,19,20,21,22]; following this, the need to develop new antimicrobial compounds continues to grow. In this scenario, the use of endogenous antimicrobial peptides (AMPs) as the latest generation of compounds to show promising antimicrobial activity is becoming increasingly fascinating. This review aims to provide a new perspective on the function of human defensins in staphylococcal cutaneous infections and their potential activities to fight *S. aureus* resistance.

## 2. *Staphylococcus aureus*: Skin Colonization and Infection

The skin is our most external barrier and is therefore exposed to a high number of pathogenic microorganisms; consequently, it provides different defense mechanisms. The skin is made up of three layers—namely, the epidermis, the dermis, and the hypodermis, starting from the outermost layer (Figure 2). The barrier that prevents the entry of microbial agents is provided by the stratum corneum (Figure 2), which represents the outermost layer of the skin, belonging to the epidermis (Figure 2). In addition, within the epidermis, it is possible to underline the presence of various immune sentinels activated only in the presence of pathogens [23,24] (Figure 2). It is important to underline that about 30% of healthy people are infected by *S. aureus* [25] through a process that depends on the conflict between the host factors and the commensal organisms delegated to contrast the colonization by the virulence factors belonging to *S. aureus*. In fact, it is the virulence factors that first cause colonization of the skin and then the subsequent infection by *S. aureus* [26]. The human skin tissue has intrinsic characteristics that prevent colonization by *S. aureus*; among these, a mention should be given to the low temperature and acidic pH [27,28]. The acidic pH is due to the presence of filaggrin. Filaggrin is an essential component of the epidermis, which, during the process of differentiation of the epidermis, is broken down into urocanic acid and carboxylic acid pyrrolidone [29]; as a consequence, the inhibition of *S. aureus* growth and expression of colonization factors such as aggregation factor B (ClfB) and fibronectin-binding protein A (FnbpA) occur [29]. In addition, the skin, on its surface, has a wide range of commensal organisms such as *Staphylococcus epidermidis*, *Propionibacterium acnes,* and the *Malassezia* species, which defend the skin from colonization and invasion by *S. aureus* and other pathogens [27,28]. Recently, *S. epidermis* has been shown to produce a serine protease, Esp, which blocks colonization of *S. aureus* by eliminating its biofilms [30]; moreover, it is capable of producing phenol-soluble modulins (PSMγ and PSMδ), which activate the toll-like 2 receptor (TLR2) on keratinocytes, further activating the production of human β-defensin 2 and 3 (hBD2 and hBD3), which exhibit a strong antimicrobial activity against *S. aureus* [31,32,33]. In this case, HBD2 and HBD3 are produced by keratinocytes and the cornea, showing a high bacteriostatic and/or bactericidal power against *S. aureus* [34,35,36,37]. In fact, it has been observed that *S. aureus* infections increase in subjects suffering from dermatitis atopic, in which there is a reduction in the levels of β-defensins [38]. The host′s defenses favor the development by *S. aureus* of various factors that promote the binding and survival of the skin surface. To bind to the host surface elements, *S. aureus* uses microbial surface components that identify adhesive matrix molecules (MSCRAMM), such as fibrinogen-binding proteins (ClfA and ClfB), fibronectin-binding protein A (Fnbp A) and Fnbp B, surface-regulated iron determinant A (IsdA), and wall-mounted teichoic acid [39,40,41,42]. Furthermore, in case of inflammation of the skin, it is possible to observe an increase of Th2 cytokines, which allows for colonization by *S. aureus* through the binding of this to fibrinogen [43]. Finally, *S. aureus* encompasses factors such as IsdA, which increase the hydrophobicity of the microorganism, countering the host′s AMP responses [44].

## 3. Methicillin-Resistant *S. aureus* (MRSA) Infection

*S. aureus* shows a marked resistance to common antibiotic therapies. The first resistant *S. aureus* strains were found a few years after the introduction of penicillin; in this case, *S. aureus* produced the β-lactamase enzyme by evading drug therapy. In 1959, methicillin was introduced; as a consequence, in 1960 [45], the first strains resistant to this antibiotic (MRSA) were highlighted. The ability to escape treatment by methicillin results from three different mechanisms (Figure 3): The first mechanism produces a penicillin-binding protein called PBP2a, encoded by the mecA gene, causing a decrease in the antimicrobial activity of β-lactams (Figure 3) [46]. Recent studies have highlighted that the new homologues of mecA genes—for example, mecB, mecC, and mecD—are probably not detectable with current laboratory methods [47,48,49]. Borderline oxacillin-resistant *S. aureus* (BORSA) represents the second process of antimicrobial evasion (Figure 3), in which there is an increase in the production of β-lactamase responsible for resistance to oxacillin. The last mechanism is highlighted in modified *S. aureus* (MODSA) (Figure 3); in particular, there is an accumulation of mutations in the transpeptidase domains in their native PBPs [50], causing resistance to methicillin. Several MRSA clones, such as health-associated MRSA (HA-MRSA) [51], community-associated MRSA (CA-MRSA) [52], and livestock-associated MRSA (LA-MRSA) [53], have emerged from modern research.

Furthermore, there are molecular differences between the CA-MRSA and HA-MRSA strains regarding the types of SCCmec; in fact, the HA-MRSA strains carry the large staphylococcal chromosome cassette (SCCmec) belonging to types I–III and containing the mecA gene, generally universal among MRSA isolates. These are generally resistant to several classes of non-β-lactam antibiotics. The large SCCmec I–III types appear to be present in the HA-MRSA strains and are transported to *S. aureus* by a commensal staphylococcal species [54]. Transporting the psm-mec locus from SCCmec type II elements reduces virulence, eliminates colony diffusion activity, decreases the expression of chromosomally encoded PSMa, and improves biofilm formation [55]. Moreover, HA-MRSA strains rarely carry Panton–Valentine leukocidin (PVL) genes. CA-MRSA isolates carry smaller SCCmec elements, commonly referred to as SCCmec type IV or type V [56]. CA-MRSA strains show resistance to fewer classes of non-β-lactam antibiotics, and are often carriers of PVL genes. Positive strains of CA-MRSA, mecA, and PBP2a have also been found, which are phenotypically sensitive to oxacillin [57]. It is assumed that the expression of mecA alone cannot be indispensable for guaranteeing the phenotypic resistance of methicillin as well as oxacillin. It is important to underline that the presence of the two-component vraS/vraR regulation system is indispensable for resistance to oxacillin in CA-MRSA [58,59]. In addition, several LA-MARSA clones can be cataloged: CC9 frequently found among cattle in Asia [60], ST5 isolated from pigs in the United States [61], and CC1 mainly expressed in hospital infections in Romania, which has a lower pathogenic activity in patients [61]. However, the manifestation of *S. aureus* clones capable of adapting and evolving rapidly has resulted in the appearance of antibiotic resistance, as well as in an increase in virulence and/or transmissibility.

## 4. Effective Human Defensins Versus *S. aureus*

Antimicrobial peptides (AMPs) are a group of small cationic molecules with antimicrobial activity (29–34 amino acids), which show a fundamental action in innate immunity against bacteria, fungi, and viruses [62,63,64]. Being cationic molecules, they have the ability to intercalate in the anionic membrane of microbes, thus causing osmotic lysis; consequently, the autolytic enzymes stimulated by AMPs correlate with bacterial cell death [65]. Commonly, AMPs are present in the skin tissue because they are produced directly by keratinocytes; on the other hand, during inflammatory processes, there is an overproduction of these to defend the skin against infections [66,67] (Table 1). Human defensins (human neutrophil peptides (HNPs)) comprise both α-defensins and β-defensins (HBD1–4) [62,63,64]. HNPs exhibit bactericidal activity against *S. aureus*, but at the same time they influence the activation of macrophages, T cells, and mast cells through a protein kinase c (PKC)-dependent mechanism [68,69,70], calling them back to the infection site. Therefore, in this review, we highlight the bacteriostatic and bactericidal abilities of the human defensins expressed in the cells of the skin tissue and in the immunity cells, used to defend the host against *S. aureus* infections [71,72].

### 4.1. Alfa-Defensins

Within the neutrophil granules, it is possible to observe a high presence of HNP1-3 and a more moderate HNP4 expression [73]. In this case, HNP2 shows a powerful bactericidal activity against *S. aureus*; meanwhile HNP1, 3, and 4 show a moderate bactericidal action against the same pathogenic microorganism [74].

### 4.2. Beta-Defensins

β-defensin represents the majority of human AMPs. These molecules have a beta sheet, conformation, and are classified according to the number and position of their disulfide bridges [75]. There are four human beta-defensins (HBD1–4) that are presented by epithelial cells, such as keratinocytes, activated monocytes/macrophages, and dendritic cells [63,64]. HBD1 is constitutively activated, while HBD2 and HBD3 are inducible in the presence of infections or cytokines [76]. Importantly, HBD1 does not show an antimicrobial activity against *S. aureus*, while in vitro experiments have shown that HBD2 and HBD4 show a weary bacteriostatic activity against *S. aureus* [77,78]. In contrast, recent in vitro and ex vivo experiments have shed light on the potent bactericidal activity of HBD3 against *S. aureus* [79,80]. In addition, in keratinocytes, the production of HBD2 and HBD3 can be stimulated by live *S. aureus* or killed by heat or bacterial elements, as well as by lipopeptides or lipoteic acid [81,82,83,84,85]. In fact, it is possible to verify that in the development of skin lesions, activation of the epidermal growth factor receptor (EGFR) occurs, which stimulates an overproduction of HBD3, providing an additional defense mechanism against *S. aureus* [66,86]. Ultimately, defensins cause the production of cytokines, including IL-8, which have a chemotactic activity [87], representing an alarm bell in inflammatory processes including bacterial infections.

## 5. Antimicrobial Barrier of the Skin 

The skin is our last barrier and is therefore exposed to a large number of different pathogens [88,89]. Keratinocytes represent the type of cells mainly expressed in the epidermis and are sentinels that contribute materially to the protection against bacterial skin infections. Keratinocytes express several Toll-like receptors (TLRs) and oligomerization domain proteins 2 (NOD2) that bind nucleotides and are crucial for identifying pathogen-associated molecular patterns (PAMPs). The identification of PAMPs causes the activation of innate immunity, which implies the secretion of different cytokines and chemokines, but also AMPs that allow immune cells to be enrolled in the area of the infection [90,91]. Although the activation of TLRs in the skin is necessary for the activation of innate and subsequently adaptive immunity, excessive stimulation can induce an uncontrollable inflammatory response that could cause inflammatory skin diseases, autoimmune diseases, or even sepsis [92]. TLRs are also present in the dermal fibroblasts and Langerhans cells of the epidermis, as well as in immune cells. Native skin cells exhibit separate TLR expression patterns and can therefore identify distinct PAMPs to trigger different host defense mechanisms [93].

## 6. Human Defensins and *S. aureus*-Dependent Skin Diseases

AMPs play a key role in the pathogenesis of cutaneous diseases, for example, psoriasis or atopic dermatitis (AD). Since AMPs are under-regulated in the damaged skin of patients with AD, these patients are more susceptible to *S. aureus* cutaneous infections. AMPs are instead increased in psoriasis and rosacea, where they aggravate inflammation related to the severity of the disease. HBD2–4 are increased in diabetic foot ulcers, but are unable to regulate wound healing and infections [94]. HBD2 and HBD3 induce the activation of pDC in psoriasis [95]. HBD2 and 3 are downregulated in lesions of atopic dermatitis (AD) [96]. The decrease of AMPs in AD could be caused by the activation of cytokines derived from Th2 IL-4, IL-10, and IL-13, which, in effect, repress the induction of AMPs. Therefore, therapies that target AMPs could improve the innate immune system and could lead to a decreased inflammation of the skin. Omiganan (a synthetic cationic indolicidin derivative), Lytixar (a synthetic cationic tripeptide), and DPK 060 (a kininogen-derived cationic random coil peptide) are the AMPs that have currently completed phase II for the evaluation of pharmacodynamics, safety, and efficacy in adults with moderate-to-severe AD (NCT02456480, NCT03091426, NCT01223222, NCT01522391); however, the results have not yet been reported. Although the efficacy of AMP therapy in AD has yet to be determined, it may have a role in AD, particularly in those with frequent or debilitating superimposed infections [97].

AMPs also appear to have a pathogenic function in AD, as they favor the production of Th2-derived cytokines implicated in the pathogenesis of AD [98]. In addition, it has been hypothesized that HBD3 has a skin barrier function, which is damaged in AD [99]. Different AMPs are over-regulated in acne vulgaris, possibly through the induction of *Propionibacterium acnes*. This upregulation of AMPs could reduce the inflammatory response due to the proinflammatory functions of AMPs, or could even be healthy because of the anti-inflammatory and antimicrobial effects [100,101]. Among the various β-defensins, the one with the greatest antimicrobial activity against *S. aureus* is HBD3, followed by HBD2 and then HBD1 [102,103,104].

Keratinocytes have the intrinsic ability to kill *S. aureus*, a phenomenon dependent on HBD3 [105]. Comparisons of skin biopsies show that HBD3 levels are similar between healthy individuals and those with AD. However, AD patients have a reduced ability to mobilize HBD3 against bacteria due to the presence of Th2 cytokines [106]. Furthermore, a higher induction of HBD3, but not of HBD2, is associated with a clinical course and better outcomes of *S. aureus* skin infections [107,108]. The level of expression of defensins can be adjusted at several stages. The genes that code for β-defensins are grouped on chromosome 8p23.1, which is a frequent site of genetic rearrangement, and therefore the number of copies of defensin genes can vary between individuals [109,110,111]. Both α- and β-defensins are secreted as proproteins, and post-translational processing is necessary to create mature defensins [112].

## 7. Common *S. aureus* Cutaneous Infections

Skin and soft tissue infections (SSTIs) are the most common bacterial diseases in people [113,114]. Bacterial SSTIs can range from superficial infections to complicated diseases that can lead to the development of sepsis with fatal results [115]. *S. aureus* is the primary cause of delayed healing and infection in both acute and chronic wounds. In addition, skin infections caused by this microorganism often produce invasive infections that can even cause sepsis [116,117]. Although *S. aureus* represents the most common source of SSTIs worldwide, followed by β-hemolytic streptococci, *Escherichia coli,* and *Pseudomonas aeruginosa*, chronic or postoperative wounds are, in most cases, caused by Gram-negative bacteria such as *P. aeruginosa*, *Enterococcus,* and *Acinetobacter* species [118,119,120]. In addition, there is a worldwide increase in community-acquired MRSA [121]; in particular, it is the inflammatory response following a bacterial infection that determines the clinical severity of *S. aureus* skin infections, rather than the bacterial load [122].

## 8. Host Defense Mediated by Human Defenses against *S. aureus* Skin Disease: The Role of IL-1 and IL-17

During skin infection, on the one hand keratinocytes produce and release IL-1α, and on the other hand macrophages and dendritic cells produce IL-1β. These two cytokines are both involved in the activation of the nuclear factor kappa-light-chain-enhancer of activated b cells (NF-κB); as a consequence, an increase in the production of HBD2 and 3 is observed. At the same time, IL-1α stimulates the production of pro-inflammatory cytokines, chemokines, and adhesion molecules that recall neutrophils from circulation to areas of skin infected with *S. aureus* (Figure 4) [121]. In addition, activating TLR2 activates IL-17, which, in turn, plays a crucial role in attracting T cells and natural killer cells (Th17, NKT) to infected areas. IL-17 activation promotes and increases the expression of HBD2 and 3 by keratinocytes and guides the recruitment of neutrophils by induction of different chemokines (CXCL1, CXCL2, and IL-8) and granulopoiesis factors (G-CSF and GM-CSF) [122].

## 9. Mechanisms by Which *S. aureus* Escapes the Human Defenses Derived from the Skin

The importance of human defensins in the host’s skin defense during *S. aureus* infection is highlighted by the mechanisms that this microorganism has developed in order to resist and escape these peptides. In fact, *S. aureus* blocks human defensins through various mechanisms.

### 9.1. Secretion of Molecules Binding Extracellular Defensins

*S. aureus* produces staphylokininase (SAK), which binds and blocks human α-defensins. As a consequence, *S. aureus* generates resistance against the α-defensins; in fact, in vitro studies have shown that SAK levels are inversely related to the susceptibility of *S. aureus* strains to α-defensins (Figure 5) [123].

### 9.2. Reduction of Net Negative Charges on the Bacterial Surface

#### 9.2.1. *S. aureus*’ Alteration of the Cell Membrane with L-lysine

*S. aureus* is able to reduce the anionic charge of the cell membrane through the addition of L-lysine on the phosphatidylglycerol residues [124]. This mechanism, called lysinylation, is made possible by a membrane protein named multiple protein of the peptide resistance factor (MprF) [125,126], which neutralizes the bacterial cell wall and reduces the sensitivity to α-defensins (Figure 5).

#### 9.2.2. *S. aureus*’ Alteration of the Cell Membrane with D-Alanine

*S. aureus* is able to alter the cell membrane through the dltABCD operon cages that express positively charged D-alanine residues to negatively charged teicoich acid residues. This mechanism generates a lower sensitivity to α-defensins [127] and, at the same time, causes a partial neutralization of the polymer [128], which reduces the interaction of the AMPs with the bacterial surface [129]. In fact, recent studies have shown that a D-alanine-free *S. aureus* mutant (dltA mutant), compared to clinical isolates that expressed lower dltA levels, showed a higher susceptibility to HBD2 and HBD3 (Figure 5) [130].

### 9.3. Modification of Skin Hydrophobicity

*S. aureus* produces IsdA in hordes to modify the skin’s hydrophobicity. In fact, human cutaneous tissue is rich in antimicrobial fatty acids produced by the sebaceous glands, which, as a consequence, prevents the entry of cell fatty acids [126]. Therefore, IsdA makes *S. aureus* refractory to HBD2 on human skin (Figure 5) [126].

## 10. Conclusions and Future Perspective

*S. aureus* is normally present in human cutaneous tissue as part of the skin microbiota. However, in specific conditions, this pathogenic microorganism is able to colonize and infect human skin tissue, destabilizing the host’s immune defenses. Recent studies have highlighted the emergence of resistance mechanisms against AMP by *S. aureus* [129]; however, there is a lot of research aimed at improving the resistance of AMPs, to use these as “warriors” against *S. aureus* infections. The mechanisms that aim to improve this resistance are varied: The first consists in the stabilization of the disulfide bridges that are intrinsic structural feature of the defensins, thus increasing the resistance to bacterial proteases [131]. The second consists in a variation of the amino acid sequence of AMPs, in which case the number of cationic amino acids is increased, generating an increase in the net positive charge, thus leading to an increase in the anti-microbial activity [131]. The third comprises the neutralization of the activity of MprF; as a consequence, there is a decrease in the infectious activity by *S. aureus*, making it more accessible to treatment with lipopeptide antibiotic daptomycin [131]. In addition, cyclization and analog engineering increase the stability of different AMPs enclosing human defensins [131,132,133,134,135,136,137,138,139,140,141]. This process improves their therapeutic efficacy; as a result, a greater susceptibility by *S. aureus* is observed, thus causing the disappearance of skin infections. The increase in virulent and resistant strains of *S. aureus* encourages the scientific community to develop new therapeutic strategies to be used in combination with the therapies currently in use. In conclusion, the continuous evolution of drugs aimed at decreasing the resistance of the pathogen, such as modified AMPs, are an excellent springboard toward the fight against *S. aureus*.

## Figures and Tables

**Figure 1 antibiotics-09-00198-f001:**
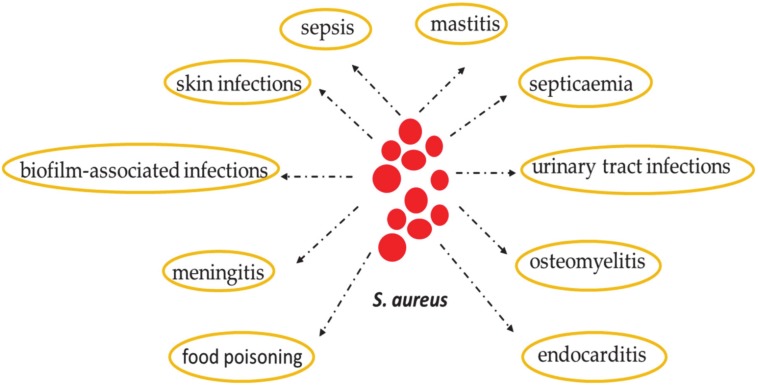
*Staphylococcus aureus* infections.

**Figure 2 antibiotics-09-00198-f002:**
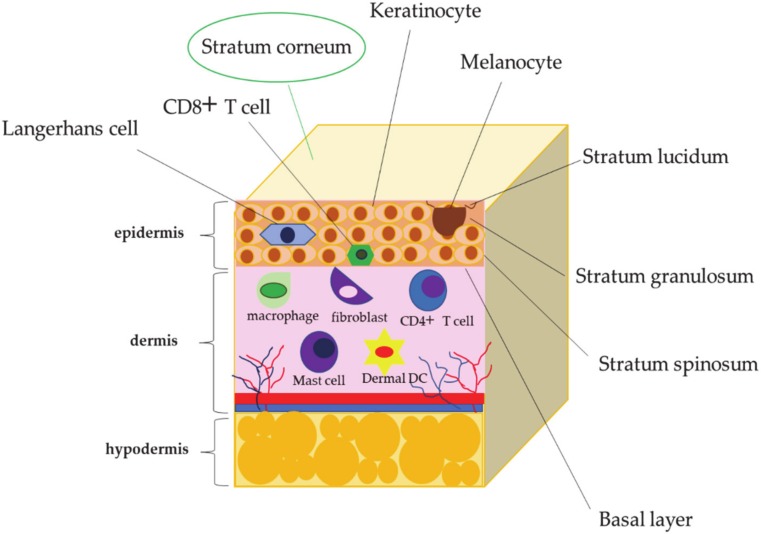
Skin organization.

**Figure 3 antibiotics-09-00198-f003:**
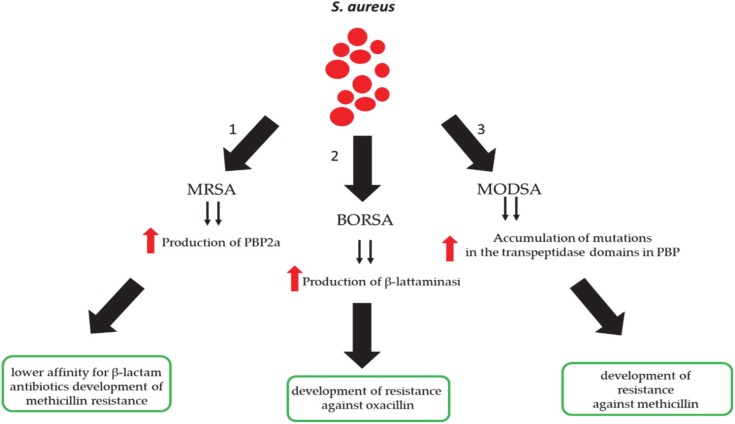
Resistance mechanisms of *S. aureus* against penicillin. MRSA, methicillin-resistant *S. aureus*; BORSA, borderline oxacillin-resistant *S. aureus*; MODSA, modified *S. aureus*.

**Figure 4 antibiotics-09-00198-f004:**
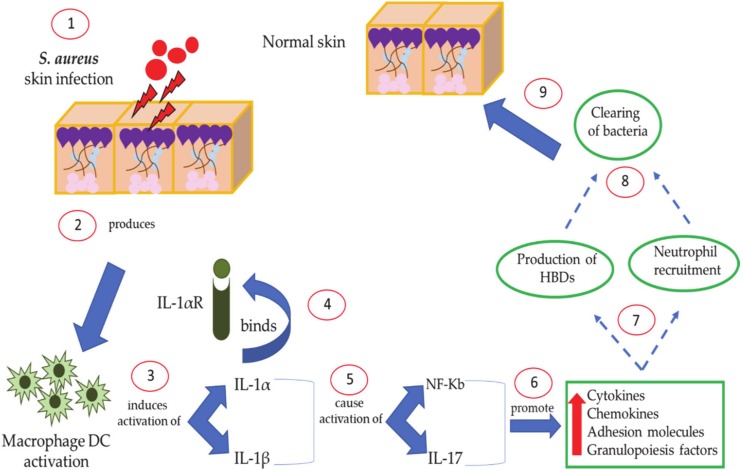
Host defense-mediated human defensins activity against *S. aureus* skin disease.

**Figure 5 antibiotics-09-00198-f005:**
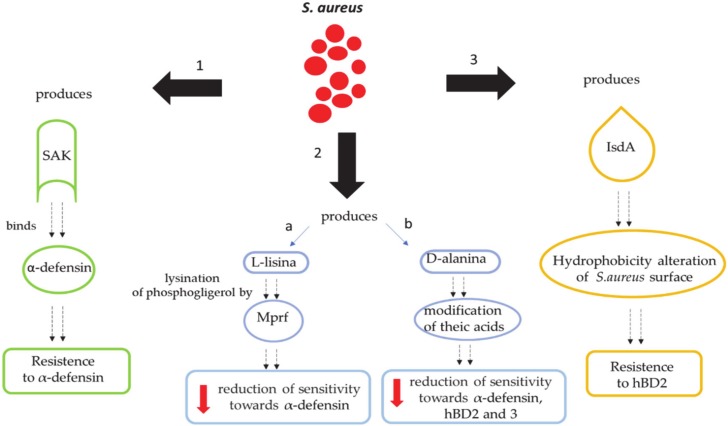
Evasion mechanisms developed by *S. aureus*.

**Table 1 antibiotics-09-00198-t001:** Human defensins involved in human skin immune defenses against *S. aureus*.

Defensin Type	Cellular Skin Production	Mechanism of *S. aureus* Evasion	References Doi Number
α-Defensins	Neutrophils	Staphylokinase, MprF, dltABCD operon	[73,74,75,76]
HBD2	Keratinocytes, macrophages, and dendritic cells	IsdA, dltABCD	[77,78,79]
HBD3	Keratinocytes	dltABCD operon	[79,80,81,82,83,84,85]
HBD4	Keratinocytes	SAEC 6043	[79]

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
