# Peer review of "Human Defensins: A Novel Approach in the Fight against Skin Colonizing *Staphylococcus aureus"

_antibiotics, 2020, doi:10.3390/antibiotics9040198_

Round 1

Reviewer 1 Report

The manuscript fits the scope of the journal. It summarizes important findings on interactions between human defensins and Staphylococcus aureus infections of the human skin. – It is well written and only requires minor revision before its acceptance for publication.

Abstract:

Line 26, Additionally to these, we will

Introduction

Line 36, bacterium. It shows opportunistic

Line 39, and infections

Line 40, and it colonizes

Lines 41-42, meaning of sentence is unclear. Staph. aureus releases SEA into food, and this enterotoxin causes food-borne poisoning.

Line 55, it is necessary to improve biosecurity

Line 134, guest?

Line 143, Consequently,

Line 151, we are highlighting

Line 160, have a beta sheet

Line 170, lesions

Line 178, protection

Line 252, is made possible by

Line 274, resistance of AMP

Lines 281-284, split into two sentences, as otherwise difficult to read

References

Please check journal formatting rule: paper tile not in italic; but journal abbreviation should be in italic

Author Response

Response to Reviewer Comments

Point: The manuscript fits the scope of the journal. It summarizes important findings on interactions between human defensins and Staphylococcus aureus infections of the human skin. – It is well written and only requires minor revision before its acceptance for publication

Response: Thanks to Reviewer 1 for the helpful comments. To satisfy your request, we have modified the text: abstract, Line 26; introduction, Line 36, Line 39, Line 40, Lines 41-42, Line 55, Line 134, Line 143, Line 151, Line 160, Line 170, Line 178, Line 252, Line 274, Lines 281-284. Additional to this, we checked the journal formatting rule for the references.

Reviewer 2 Report

In the review “Human Defensins: a novel approach in the fight of skin colonizing Staphylococcus aureus”, authors Scudiero et al have reviewed the role of S. aureus in regards to the host-pathogen interactions in skin disorders and acquired resistance of S. aureus to methicillin (MRSA). In addition, they have also highlighted the role of defensins and other innate immune mediators like toll-like receptors, interleukin 1 and interleukin 17.  The articles seem nailing very important aspect of S. aureus infection and antibiotic resistance associated with it.

Major concerns:

My first impression/concern about the review article is a poor use of the English language, I can write out pages pointing to the inappropriate use of the English language. The language is very non-scientific. However, I would just say extensive correction of language and grammar is required.

The review seems to be very similar to the 2014 IJMS paper, Ryu, S.; Song, P. I.; Seo, C. H.; Cheong, H.; Park, Y. Colonization and infection of the skin by S. aureus: immune system evasion and the response to cationic antimicrobial peptides. Int. J. Mol. Sci. 2014, 15, 8753– 8772. Where they have reviewed all cationic antimicrobial peptides against S. aureus. Figure 2 is almost identical to figure 4 of this review article.

The overwhelming literature floating around related to cationic antimicrobial peptides and S. aureus infection, the references can be more comprehensive.   

Many statements seem to be very repetitive like the following ones:

S.aureus is the major cause of nosocomial wound infections and is responsible for delayed wound healing, prolonged inflammation, and chronic infection.

Author Response

Response to Reviewer 2 Comments

In the review “Human Defensins: a novel approach in the fight of skin colonizing Staphylococcus aureus”, authors Scudiero et al have reviewed the role of S. aureus in regards to the host-pathogen interactions in skin disorders and acquired resistance of S. aureus to methicillin (MRSA). In addition, they have also highlighted the role of defensins and other innate immune mediators like toll-like receptors, interleukin 1 and interleukin 17.  The articles seem nailing very important aspect of S. aureus infection and antibiotic resistance associated with it.

Major concerns:

My first impression/concern about the review article is a poor use of the English language, I can write out pages pointing to the inappropriate use of the English language. The language is very non-scientific. However, I would just say extensive correction of language and grammar is required.

The review seems to be very similar to the 2014 IJMS paper, Ryu, S.; Song, P. I.; Seo, C. H.; Cheong, H.; Park, Y. Colonization and infection of the skin by S. aureus: immune system evasion and the response to cationic antimicrobial peptides. Int. J. Mol. Sci. 2014, 15, 8753– 8772. Where they have reviewed all cationic antimicrobial peptides against S. aureus. Figure 2 is almost identical to figure 4 of this review article.

The overwhelming literature floating around related to cationic antimicrobial peptides and S. aureus infection, the references can be more comprehensive.   

Many statements seem to be very repetitive like the following ones:

S.aureus is the major cause of nosocomial wound infections and is responsible for delayed wound healing, prolonged inflammation, and chronic infection.

Response

Thanks to Reviewer 2 for the helpful comments. In order to satisfy our request we have modified  Figure 4, we improved the English of the Manuscript and we made some changes in the text in order not to be repeated.

Reviewer 3 Report

The proposed review article is concise and comprehensive. 

The title is informative and relevant. The references are relevant and recent. The cited sources are referenced correctly. Appropriate and key studies are included.

The introduction reveals what is already known about this topic. 

The illustrative materials are relevant and clearly presented. 

The conclusions are supported by references. The article is consistent within itself.

Author Response

Response to Reviewer 3 Comments

The proposed review article is concise and comprehensive.

The title is informative and relevant. The references are relevant and recent. The cited sources are referenced correctly. Appropriate and key studies are included.

The introduction reveals what is already known about this topic.

The illustrative materials are relevant and clearly presented.

The conclusions are supported by references. The article is consistent within itself.

Response

Thanks to Reviewer 3 for the comments.

Reviewer 4 Report

The work is very interesting, it summarizes current knowledge about human defensins and their potential application in the treatment of infection and / or colonization of S. aureus. The well-prepared figures are a special value of the work.

There are some minor issues which need improvement:

  • "S. aureus is a common component of the human microbiota" please specify what type of microbiota - skin, gut?
  • In the "Introduction", in addition to information about S. aureus infections, the negative impact of colonization on some chronic inflammatory dermatoses, in particular atopic dermatitis, should be mentioned.
  • "It is important to underline that about 30% of healthy people are infected by S. aureus" - infected or colonized?
  • Are there data on defense use in clinical trials in atopic dermatitis
  • References require an update, most of the cited publications are over 10 years old

Author Response

Response to Reviewer 4 Comments

The work is very interesting, it summarizes current knowledge about human defensins and their potential application in the treatment of infection and / or colonization of S. aureus. The well-prepared figures are a special value of the work.

There are some minor issues which need improvement:

  1. "S. aureus is a common component of the human microbiota" please specify what type of microbiota - skin, gut?
  2. In the "Introduction", in addition to information about S. aureus infections, the negative impact of colonization on some chronic inflammatory dermatoses, in particular atopic dermatitis, should be mentioned.
  3. "It is important to underline that about 30% of healthy people are infected by S. aureus" - infected or colonized?
  4. Are there data on defense use in clinical trials in atopic dermatitis
  5. References require an update, most of the cited publications are over 10 years old

Response

Thanks to Reviewer 4 for the helpful comments. In order to satisfy your request, we have modified the text, moreover, we added some specifications and references.

  1. "S. aureus is a common component of the human microbiota" please specify what type of microbiota - skin, gut? SKIN ( pag1 line 40)
  2. In the "Introduction", in addition to information about S. aureus infections, the negative impact of colonization on some chronic inflammatory dermatoses, in particular atopic dermatitis, should be mentioned. (Shi, B.; Leung, D.Y.; Taylor, P.A.; Li, H. Methicillin-resistant Staphylococcus aureus colonization is associated with decreased skin commensal bacteria in atopic dermatitis. J Invest Dermatol 2018, 138, 1668-1671 and Kim, J.; Kim, B.E.; Leung, D.Y. Pathophysiology of atopic dermatitis: clinical implications. Allergy Asthma Proc. 2019, 40, 84-92)
  3. "It is important to underline that about 30% of healthy people are infected by S. aureus" - infected or colonized? (We have changed the text page 1-2 line 40-46)
  4. Are there data on defence use in clinical trials in atopic dermatitis (We have changed the text page 6 lines 203-210 and we specified the existence of some trials)
  5. References require an update, most of the cited publications are over 10 years old. (we have added some references to update our dataset, the changes have been highlighted in yellow)

All the changes that have been suggested to us have been highlighted in yellow.

Round 2

Reviewer 2 Report

In the revised version of “Human defensins: a novel approach in the fight of skin colonizing Staphylococcus aureus”, Scudiero et al. have made very few changes in the English language the review article still needs extensive English language. The language in the review is very immature and lacks basic scientific jargons.  Most of the times statements seems to be repetitive and irrelevant.

The very first two statements of the introduction section “Staphylococcus aureus (S. aureus) is a Gram positive bacterium. Its shows opportunistic pathogen activity in both humans and animals.” – is not only wrong English but very confusing it self.   Few of the abbreviations needs to be like HA-MRSA, SCCmec is addressed first and then the abbreviation is answered. There are many such small glitches in this review article, which basically indicates that attention to the details was not been considered.  I strongly suggest the authors to get an English language check.

However, I am quite satisfied by the scientific content of the review, again we do not gain any additional information from this review which was not covered in “Ryu, S.; Song, P. I.; Seo, C. H.; Cheong, H.; Park, Y. Colonization and infection of the skin by S. aureus: immune system evasion and the response to cationic antimicrobial peptides. Int. J. Mol. Sci. 2014, 15, 8753– 8772”.

Author Response

Reviewer 2

Comments and Suggestions for Authors

In the revised version of “Human defensins: a novel approach in the fight of skin colonizing Staphylococcus aureus”, Scudiero et al. have made very few changes in the English language the review article still needs extensive English language. The language in the review is very immature and lacks basic scientific jargons. Most of the times statements seems to be repetitive and irrelevant.

Response

Thanks to the Reviewer for the helpful comments. To meet your request we have modified the text and improved English with the support of the English editing of the magazine.

The very first two statements of the introduction section “Staphylococcus aureus (S. aureus) is a Gram positive bacterium. Its shows opportunistic pathogen activity in both humans and animals.” – is not only wrong English but very confusing itself. Few of the abbreviations needs to be like HA-MRSA, SCCmec is addressed first and then the abbreviation is answered. There are many such small glitches in this review article, which basically indicates that attention to the details was not been considered. I strongly suggest the authors to get an English language check.

Response

Thanks to the Reviewer for the helpful comments. To meet your request we have modified the text and improved English with the support of the English editing of the magazine. Furthermore, whenever we have used an acronim we have specified the meaning in full, so as to make reading more fluent and pleasant.

However, I am quite satisfied by the scientific content of the review, again we do not gain any additional information from this review which was not covered in “Ryu, S.; Song, P. I.; Seo, C. H.; Cheong, H.; Park, Y. Colonization and infection of the skin by S. aureus: immune system evasion and the response to cationic antimicrobial peptides. Int. J. Mol. Sci. 2014, 15, 8753– 8772”.

Response

Thanks to the Reviewer for the helpful comments the article by Ryu et al is a 2015 article, in our case we have tried to make an update of the knowledge found so far, in particular we have also focused on the engineering use of defensins to protect man from S. aureus, since in 2015 it was not yet known.

Ref: Falanga, A.; Nigro, E.; De Biasi, M.G.; Daniele, A.; Morelli, G.; Galdiero, S.; Scudiero, O. Cyclic Peptides as Novel Therapeutic Microbicides: Engineering of Human Defensin Mimetics. Molecules. 2017, 20, pii: E121